# Antibiotic Prescribing Trends in Belgian Out-of-Hours Primary Care during the COVID-19 Pandemic: Observational Study Using Routinely Collected Health Data

**DOI:** 10.3390/antibiotics10121488

**Published:** 2021-12-04

**Authors:** Annelies Colliers, Jeroen De Man, Niels Adriaenssens, Veronique Verhoeven, Sibyl Anthierens, Hans De Loof, Hilde Philips, Samuel Coenen, Stefan Morreel

**Affiliations:** 1Department of Family Medicine & Population Health (FAMPOP), Faculty of Medicine and Health Sciences, University of Antwerp, 2610 Antwerp, Belgium; Jeroen.DeMan@uantwerpen.be (J.D.M.); niels.adriaenssens@uantwerpen.be (N.A.); Veronique.verhoeven@uantwerpen.be (V.V.); sibyl.anthierens@uantwerpen.be (S.A.); hilde.philips@uantwerpen.be (H.P.); samuel.coenen@uantwerpen.be (S.C.); stefan.morreel@uantwerpen.be (S.M.); 2Laboratory of Physiopharmacology, Department of Pharmaceutical Sciences, Faculty of Pharmaceutical, Biomedical and Veterinary Sciences, University of Antwerp, 2610 Antwerp, Belgium; hans.deloof@uantwerpen.be; 3Laboratory of Medical Microbiology, Vaccine & Infectious Disease Institute (VAXINFECTIO), Faculty of Medicine and Health Sciences, University of Antwerp, 2610 Antwerp, Belgium

**Keywords:** anti-bacterial agents, COVID-19, primary health care, out-of-hours medical care

## Abstract

Antibiotic overprescribing is one of the main drivers of the global and growing problem of antibiotic resistance, especially in primary care and for respiratory tract infections (RTIs). RTIs are the most common reason for patients to consult out-of-hours (OOH) primary care. The COVID-19 pandemic has changed the way general practitioners (GPs) work, both during office hours and OOH. In Belgian OOH primary care, remote consultations with the possibility of issuing prescriptions and telephone triage were implemented. We aimed to describe the impact of COVID-19 on GPs’ antibiotic prescribing during OOH primary care. In an observational study, using routinely collected health data from GP cooperatives (GPCs) in Flanders, we analyzed GPs’ antibiotic prescriptions in 2019 (10 GPCs) and 2020 (20 GPCs) during OOH consultations (telephone and face-to-face). We used autoregressive integrated moving average (ARIMA) modeling to identify any changes after lockdowns were implemented. In total, 388,293 contacts and 268,430 prescriptions were analyzed in detail. The number of antibiotic prescriptions per weekend, per 100,000 population was 11.47 (95% CI: 9.08–13.87) or 42.9% lower after compared to before the implementation of lockdown among all contacts. For antibiotic prescribing per contact, we found a decrease of 12.2 percentage points (95% CI: 10.6–13.7) or 56.5% among all contacts and of 5.3 percentage points (95% CI: 3.7–6.9) or 23.2% for face-to-face contacts only. The decrease in the number of prescriptions was more pronounced for cases with respiratory symptoms that corresponded with symptoms of COVID-19 and for antibiotics that are frequently prescribed for RTIs, such as amoxicillin (a decrease of 64.9%) and amoxicillin/clavulanate (a decrease of 38.1%) but did not appear for others such as nitrofurantoin. The implementation of COVID-19 lockdown measures coincided with an unprecedented drop in the number of antibiotic prescriptions, which can be explained by a decrease in face-to-face patient contacts, as well as a lower number of antibiotics prescriptions per face-to-face patient contact. The decrease was seen for antibiotics used for RTIs but not for nitrofurantoin, the first-choice antibiotic for urinary tract infections.

## 1. Introduction

The start of the COVID-19 pandemic raised new and radical logistical and clinical challenges for primary care physicians, who were forced to respond promptly to a fast-changing environment [1,2,3,4]. Out-of-hours (OOH) primary care rapidly reorganized itself. In Belgium, general practitioners (GPs) set up, for the first time ever, telephone consultations and triage for their patients, with the aim of reducing the risk of contamination [5]. This resulted in a substantial decline in the number of face-to-face consultations and home visits, partially compensated by the rise in telephone consultations [6]. GP cooperatives (GPCs) in Belgium are set up to provide care for patients within a specified region, during weekends and bank holidays. They continued this during the pandemic, adding triage and telephone consultations, in order to separate patients suspected of having COVID-19 from other patients.

Antimicrobial resistance and the widespread use of antibiotics is a major global health problem, and the antimicrobial stewardship principles remain important during this global pandemic [7]. Before the COVID-19 pandemic, infections were the number one reason to visit a GP during OOH, characterized by an overprescribing of antibiotics, in particular for respiratory tract infections (RTIs) [8,9]. During the pandemic, GPs in OOH care were still confronted with time pressure, limited diagnostic tools and patients unknown to them [10,11]. At the beginning of the pandemic, many preliminary and contradictory scientific studies were produced at a high rate and speed, regarding, amongst other issues, whether or not to use empiric antibiotics or azithromycin [12,13,14,15]. Most of these studies took place in hospital or in intensive care units [16,17,18] and, therefore, guidance was missing for GPs as to whether they should prescribe antibiotics for suspected COVID-19 cases. Early recommendations based on Chinese experience suggested a low threshold for prescribing antibiotics for home care [19]. Many of the patients hospitalized during this period were given (broad spectrum) antibiotics for suspected secondary bacterial co-infection [20]. Furthermore, data from hospitalized patients showed an increase in the use of antibiotics, often azithromycin [21,22,23]. However, the PRINCIPLE trial evaluated the use of azithromycin in primary care and did not find an effect on reducing the time to recovery or risk of hospitalization for people with suspected COVID-19 in the community [24].

Early in the pandemic, the Belgian Antibiotic Policy Coordination Committee (BAPCOC) issued a recommendation on 23 March not to prescribe azithromycin or any other antibiotic for COVID-19-suspected cases in ambulatory care and to adhere to the Belgian antibiotic guidelines if choosing an antibiotic [25]. Similarly, in the neighboring country of the Netherlands, Dutch GPs were advised to follow the current guidelines for the diagnosis and treatment of RTIs and to prescribe amoxicillin as a first-choice treatment [26]. In the WHO guideline issued 13 March 2020 there is no clear advice for mild or moderate COVID-19 and advice to only start antibiotics in cases of severe COVID-19 with sepsis [27]. Finally, the WHO guideline of 27 May 2020 advised against the use of antibiotic therapy or prophylaxis for suspected or confirmed COVID-19 patients, unless there was a clinical suspicion of a bacterial infection and not to use broad spectrum antibiotics [28].

This observational study describes trends in the antibiotic prescribing of GPs for RTIs, before and after the start of the COVID-19 lockdown in Belgian OOH care, using routinely collected health data.

## 2. Results

### 2.1. Population Characteristics

The database contained 481,362 contacts from 2019 and 2020, of which 93,531 were excluded: 36,956 COVID-19-test contacts (solely contacts for testing reasons), 57,447 home visits and 668 because of a data collection problem. We included 388,293 contacts and 268,430 prescriptions: on average 0.69 (SD 0.89) prescriptions per contact, ranging from 1 to 12 prescriptions. Patients’ average age was 34.0 years (SD 23.5) and 54.1% were female. The study population before the start of the first lockdown (13 March 2020) was older (mean 36 years) and more often female (54.3%), compared to the population after implementation of the lockdown (mean 31 years and 53.8% female).

### 2.2. Trends in Antibiotic Prescribing

#### 2.2.1. General Trend in Antibiotic Prescribing

We observed a drop (i.e., step change) of 11.47 (95% CI: 9.08–13.87) or 42.9% in the number of antibiotic prescriptions per weekend, per 100,000 inhabitants, after compared to before the implementation of the lockdown. Figure 1 shows the trend of systemic antibiotic prescriptions per 100,000 inhabitants in 2019 and 2020. We provide the same trends for possible RTI-related and RTI-unrelated contacts as the Appendix A. The described pattern is more pronounced for cases with possible COVID-19-related symptoms (most respiratory symptoms/diagnoses, excluding throat and/or ear infections).

#### 2.2.2. Trends in Prescribing of Amoxicillin and Amoxicillin/Clavulanate

Amoxicillin and amoxicillin/clavulanate are the antibiotics most often used for RTIs. Figure 2 shows the number of amoxicillin and the number of amoxicillin/clavulanate prescriptions per weekend, per 100,000 inhabitants.

The observed decrease was larger for amoxicillin than for amoxicillin/clavulanate with 7.31 (95% CI: 5.67–8.95) or 65.2% and 2.20 (95% CI: 1.74–2.67) or 38.1% fewer prescriptions, respectively. The average number of antibiotic prescriptions for the period before and after lockdown can be found in Table 1. 

#### 2.2.3. Trends in Prescribing of Nitrofurantoin

Figure 2 also shows the number of nitrofurantoin prescriptions per weekend, per 100,000 inhabitants. We did not observe a decrease in the number of prescriptions for nitrofurantoin (−0.12 (95% CI: −0.35–0.12)), which is the first-choice antibiotic for urinary tract infections (UTI).

We added the trend in UTI diagnoses per population of 100,000 and the UTI contacts, where nitrofurantoin was prescribed, per population of 100,000, which both stayed stable before and after the lockdown, as Appendix A.

#### 2.2.4. Trend in Antibiotic Prescribing per Contact Type (Face-to-Face and Face-to-Face + Telephone)

On average, the total number of contacts did increase after lockdown (Table 1). This increase can be attributed to the emerging number of telephone contacts which were implemented as common practice after lockdown. The number of face-to-face contacts, however, decreased by almost a third.

Antibiotic prescribing per contact, both face-to-face and telephone contacts, decreased by 12.2 percentage points (95% CI: 10.6–13.7) or 56.5%, while it decreased by 5.3 percentage points (3.7–6.9) or 23.2% when only taking face-to-face contacts into account. The decrease in the total number of antibiotics prescribed during all face-to-face contacts could be attributed to the decrease of 42.8% for antibiotics prescribed per face-to-face contact and to the decrease of 57.2% for the number of contacts. Figure 3 shows the proportion of prescriptions per face-to-face contact, per weekend, before and after the lockdown.

## 3. Discussion

Our findings show a remarkable decrease in the number of antibiotic prescriptions issued in OOH primary care after the implementation of the COVID-19 pandemic lockdown. When focusing on RTI-related contacts (excluding ear/throat infections), this decline is even more pronounced. We see a marked decrease in antibiotic prescriptions for RTI-related face-to-face contacts after the start of the first wave, only, for a small part, compensated by antibiotic prescriptions issued during telephone consultations. Only a limited number of antibiotic prescriptions were delivered during telephone contacts, which was a new service implemented after the start of the lockdown in 2020. In the summer of 2020, we see a partial recovery of antibiotic prescribing during OOH care, followed by a second decline during the second wave.

The observed decrease in antibiotic prescribing during OOH care at the population level can be explained by both a decrease in the number of face-to-face contacts and a lower number of prescriptions per face-to-face patient contact, before versus after the implementation of lockdown.

The observed decline in the number of antibiotic prescriptions after the start of the pandemic is even more pronounced when looking at the number of prescriptions for amoxicillin. Amoxicillin is the most commonly prescribed and first-choice antibiotic for most RTIs.

Throughout Europe, primary care reorganized itself to provide primary care for patients with or without COVID-19 [1,2,3,4]. In this rapidly evolving pandemic situation, primary care plays a vital role in the health system [30]. GPs in the OOH primary care setting worked in a fast-changing new setting and were suddenly faced with triage and clinical decision making in very uncertain times and the treatment of possible coronavirus infections.

During the first wave of the pandemic, there was no access to COVID-19 tests. Later, COVID-19 tests were available, but even then, the results would not yet be available during the patient contact [31]. Therefore, it remained difficult for GPs to differentiate between COVID-19, bacterial (co-)infection and other diagnoses, such as common viral RTIs.

Patients’ threshold to consult their GP was potentially higher and reasons could be: the implementation of triage, the discouragement of patients to contact healthcare unless it was urgent, the discouragement of non-essential movements, the fear of patients to have a healthcare contact or the communicated pressure on healthcare by the media [2,6,32,33]. GPs felt that they had changed their clinical decision making and largely focused on respiratory assessment and triage [1]. In line with our findings, a review by Moynihan et al. showed that healthcare use, including primary and hospital care, decreased by about a third during the pandemic, mostly for patients with less severe illnesses [34].

In primary care, findings are mixed, but in general, they indicate a decrease when it comes to the number of antibiotic prescriptions dispended [4]. For example, during the beginning of the COVID-19 pandemic, the number of outpatients with antibiotic prescriptions decreased substantially in the United States [35]. A study in Iceland showed that there was an increase in the total number of medication prescriptions; however, the number of antibiotic prescriptions remained stable [36]. English data from primary care showed that the prescribing of antibiotics decreased significantly [37,38] and also, data specifically from OOH care showed a reduction in antibiotic prescribing during the first wave of COVID-19 [39]. However, when looking at the number of prescriptions per contact, during the first months of the pandemic, there is a significant increase, possibly explained by an increase in inappropriate antibiotic use in telephone consultations [40]. In the Netherlands, there was a sharp drop in the total number of antibiotic prescriptions in primary care during the first months of the COVID-19 pandemic. Possible explanations for this decrease in antibiotic prescriptions are: fewer consultations due to the discouragement of patients to contact a GP for mild RTIs, more telephone consultations and consultations in first line hubs (regionally centralized care for patients with RTI symptoms), hygiene measures and social distancing measures [41]. Except for consultations in first line hubs, which were not implemented in Belgium, similar reasons may play a role, in addition to the fact that Belgian patients were not yet acquainted with telephone consultations or triage before the COVID-19 pandemic, possibly making the threshold to consult higher. Further qualitative research could give more insight in why GPs’ prescribing behavior decreased.

With all the challenges evoked by the outbreak of COVID-19, one must not forget the challenge of antibiotic resistance [42,43]. Bacterial co-infection is uncommon in patients admitted to hospital with community-acquired COVID-19; therefore, prudent use of antibiotics remains important, even when suspecting a COVID-19 infection [22]. Worldwide antimicrobial stewardship programs and antimicrobial resistance surveillance has been impacted [44].

### Strengths and Limitations

A strength of our study is the use of interrupted time series (ITS), which provides one of the strongest evaluative designs if randomization is not possible. Moreover, as our study was undertaken in a real-world setting, the use of ITS results in strong external validity [45]. The use of routinely collected data ensures a high grade of completeness of data.

We did not assess the appropriateness of prescribing, and it is unclear whether there was, for example, an undertreatment of infections or a rise in complications. Data were routinely registered by GPs and automatically extracted from the electronic health record. Although registration is required, we have to take into account a possible under-registration or incorrect registration by the GPs. It is not possible to document a shift of prescribing from OOH to in-hours with our current data set. Belgian pharmacy data show that there was an overall significant decrease in antibiotic prescribing in Belgium, but not for chronic medications [46].

There was a lower age of patients presenting after the start of the lockdown. We did not compare the same age groups pre- and post-COVID-19; therefore we do not know what percentage of the difference can be explained by the difference in age.

We were not able to control for potential seasonal effects, as data were only analyzed for one time unit (i.e., 1 year) preceding the event. However, no clear patterns of seasonality could be distinguished from the graphical presentation of the data.

## 4. Materials and Methods

### 4.1. Study Design and Setting

We performed an observational study using health data routinely collected in 20 GPCs (10 in 2019, 20 in 2020), covering a population of 3,162,345 (1,914,541 in 2019), in rural as well as urban areas, situated in the northern Dutch-speaking part of Belgium (Flanders).

In Belgium, the first wave of the COVID-19 pandemic occurred between 1 March and 21 June 2020 (with a peak in week 15) and the second wave between 31 August 2020 and 14 February 2021 (with a peak in weeks 45–46). The first lockdown was a strict lockdown with, amongst other restrictions, closure of all schools, closure of non-essential shops, a travel ban and mandatory teleworking, while for example, during the second lockdown, school contacts were only partially limited, travelling was limited, non-essential shops were closed again and teleworking was encouraged when possible.

### 4.2. Data Sources

All variables were collected using routinely collected electronic health record data, extracted from the iCAREdata database [47,48].

### 4.3. Studied Contacts

All contacts with a GP, for patients with a national insurance number, presenting themselves at a participating GPC during weekends and bank holidays (from here onwards we refer to weekends and bank holidays as weekends) in 2019 and 2020 were studied. Home visits were excluded due to a lack of correct prescription registration. Starting in May 2020, most of the studied GPCs were offered a COVID-19 testing service by a nurse. These patients were either not ill (travel testing and contact tracing) or had a previous telephone consultation (the government did not recommend a face-to-face consultation). These testing contacts were excluded. Due to a data collection problem, the contacts during the weekend of 18 September 2020 were excluded as well.

The units of analysis were contacts and prescriptions. Our study population corresponds with the Flemish (Dutch speaking part of Belgium) population for age and gender.

### 4.4. Variables

The following variables were collected: timing of presentation (date and time), contact type (telephone consultation, face-to-face consultation or home visit), sex (male or female), age (years), diagnosis (one ICPC-2 code, International Classification of Primary Care) [49] and prescriptions (one or more Anatomical Therapeutic Chemical (ATC)-codes) [29].

The following outcomes were studied: (1a) the number of systemic antibiotic prescriptions (ATC chapter J01) per weekend, per 100,000 inhabitants, for 2019 and 2020, (1b) the number of antibiotic prescriptions (ATC chapter J) per weekend, per 100,000 inhabitants, for 2019 and 2020, for RTI-unrelated contacts and for RTI-related contacts (respiratory symptoms that correspond with symptoms of COVID-19, defined as ICPC codes: “R74: acute upper respiratory tract infection”, “R83: other airway infections”, “R81: pneumonia”, “A77: other viral infections”, “A78: other infections”, “A03: fever”, “R02: dyspnoea”, “R05: coughing”, “R80: influenza” and “R78: acute bronchitis”), (2) the number of amoxicillin and amoxicillin/clavulanate prescriptions per weekend, per 100,000 inhabitants (representing antibiotic use for RTIs, since amoxicillin and amoxicillin/clavulanate are the most commonly prescribed antibiotics for RTIs), (3) the number of nitrofurantoin prescriptions per weekend, per 100,000 inhabitants (representing antibiotic use for non-respiratory infections) and (4) the number of antibiotic prescriptions per weekend, per number of face-to-face and telephone contacts.

### 4.5. Data Analysis

The outcome variables were rates of the number of prescriptions per weekend, per 100,000 population, for different groups of antibiotics: J01, amoxicillin (J01CA04), amoxicillin/clavulanate (J01CR02) and nitrofurantoin (J01XE01). Long weekends were readjusted based on the number of hours (84 vs. 60). Interrupted time series (ITS) analysis was used to quantify changes between the period before lockdown was implemented (i.e., 1 January 2019 to 12 March 2020) and the period after implementation (i.e., 13 March 2020 to 31 December 2020), while accounting for autoregression and controlling for age and gender. Auto-Regressive Integrated Moving Average (ARIMA) models were used as they accommodate continuous variables, such as the rates we used as outcome variables [50]. To identify the most appropriate ARIMA model, a variation of the Hyndman–Khandakar algorithm was used, which iteratively searches based on a minimization of criteria such as the Akaike information criterion and maximum likelihood estimation [51]. The algorithm is available from the “forecast package” in R. We assumed the impact of the restrictions would appear as an immediate drop or ‘step’ change in the outcome variables. To account for this change, a binary variable was included in the model, indicating pre- and post-lockdown. After application of the model, residuals did not show any obvious pattern of autocorrelation and approached a normal distribution. More details on the selected models can be found in the Appendix A.

## 5. Conclusions

The COVID-19 pandemic has led to an unprecedented drop in the number of antibiotic prescriptions in Belgian OOH primary care after the implementation of the lockdown, which can be explained by a decrease in the number of face-to-face contacts as well as lower antibiotic prescribing rates per face-to-face contacts. This result was seen for antibiotics used for RTIs but not for nitrofurantoin, the first-choice antibiotic for UTI.

## Figures and Tables

**Figure 1 antibiotics-10-01488-f001:**
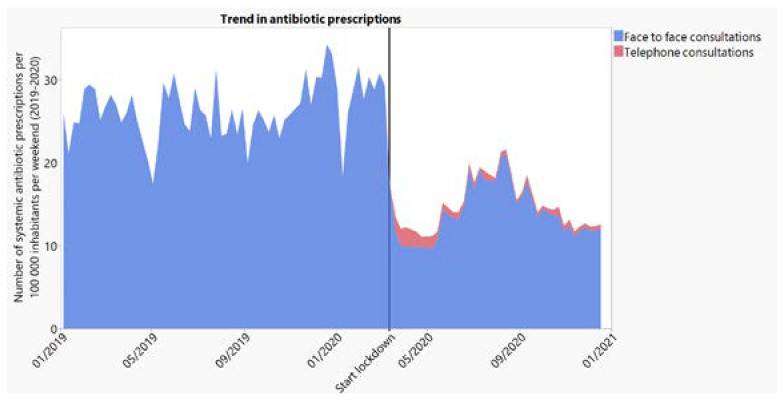
Number of systemic antibiotic prescriptions per 100,000 inhabitants, per weekend over time (2019–2020).

**Figure 2 antibiotics-10-01488-f002:**
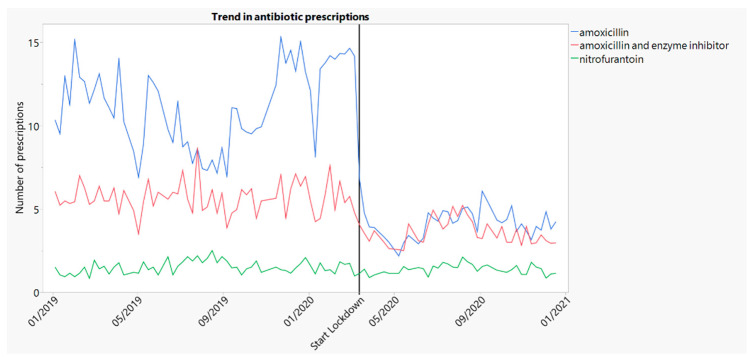
Number of amoxicillin, amoxicillin/clavulanate and nitrofurantoin prescriptions per 100,000 inhabitants, per weekend over time (2019–2020).

**Figure 3 antibiotics-10-01488-f003:**
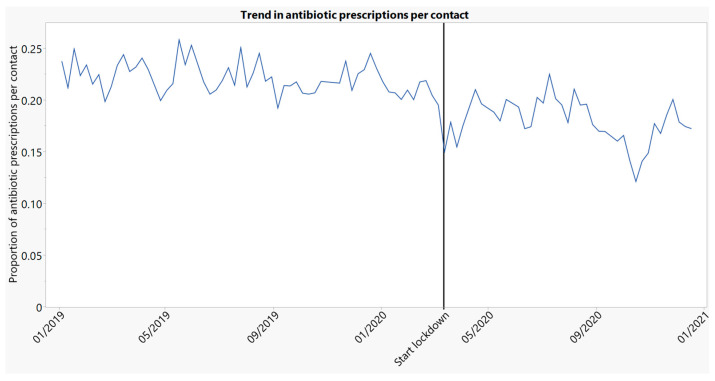
Proportion of prescriptions per face-to-face contact, per weekend over time (2019–2020).

**Table 1 antibiotics-10-01488-t001:** Before and after lockdown comparison of mean number of contacts per type of contact and mean number of antibiotic prescriptions per type of antibiotic.

	Before Lockdown	After Lockdown
Number of contacts per weekend, per 100,000 population (SD)		
- All	121 (15)	160 (39)
- Face-to-face	120 (15)	85 (17)
- Telephone	1 (0)	75 (29)
Number of antibiotic prescriptions per weekend, per 100,000 population (SD)		
- All (J01)	26 (3)	15 (3)
- Amoxicillin (J01CA04)	11 (2)	4 (1)
- Amoxicillin/clavulanate (J01CR02)	6 (1)	4 (1)

SD: standard deviation; J01, J01CA04, J01CR02: Anatomical Therapeutic Chemical codes [29].

## Data Availability

Given the privacy policy of the iCAREdata database, the authors are not allowed to share their entire database. Sharing this database would potentially harm the privacy of the included patients as one might get information about their identity by combining data from several columns (variables). We are, however, able to deliver a selection of columns upon reasonable request. A part of the iCAREdata database is disclosed to the public on a website (https://icare.uantwerpen.be accessed on 3 December 2021). This includes the data presented in this article but with less detail. Access to our data can be requested by contacting icaredata@uantwerpen.be.

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
