# Peer review of "Antibiotic Prescribing Trends in Belgian Out-of-Hours Primary Care during the COVID-19 Pandemic: Observational Study Using Routinely Collected Health Data"

_antibiotics, 2021, doi:10.3390/antibiotics10121488_

Round 1
Reviewer 1 Report
The present paper presents the course of antibiotic prescribing over time in OOH GP-care in Belgium (2019-2020). Changes in numbers of prescriptions (total, amox, augm) are presented. The trend in prescription before-after lockdown is analysed using ITS. Data for nitrofurantion is analysed separately. A large decrease in prescription after lockdown is found.
The manuscript is clear, relevant for the field and presented in a well-structured manner.
The study design is appropriate.
General:
It would be valuable to see not only the numbers of antibiotic prescriptions but also the course of the number of contacts. These numbers are needed to support the conclusion of the paper (the COVID-19 pandemic has led to an unprecedented drop in the number of antibiotic prescriptions in Belgian OOH primary care, which is mostly explained by a decrease in the number of patient contacts after the implementation of the lockdown)
With respect to the second part of the conclusion: GPs also prescribed fewer antibiotics per face-to-face patient contact, but this decrease was much
smaller: I cannot find the decrease over time for TCs in a figure. And how can GPs prescribe less than before COVID if before COVID there was no option for TC?
Did you look at the course of contacts for complications before/after lockdown? (pneumonia/mastoiditis/pyelonefritis). Do we have an idea whether the decrease in AB was safe?
Introduction:
Were there differences in preventive measures between the first and second wave? Does this help explain the difference in decrease first/second wave?
Results
Could data be added on UTI contacts/nitrofurantoin per UTI contact? Nitro- numbers stayed the same: did number of UTI contact go down? And the prescription rate go up?
The figures show a clear picture over time. The numbers in the figures are highly variable over time. Could a line with a mean before/after lockdown be added. So the overall decline numbers can be visualised?
i.e. Figure three: text: Antibiotic prescribing per contact, both face-to-face and telephone contacts, de creased by 12.0% ( 95% CI: 10.8 – 13.3%) while it decreased by 5.2 % (3.8 – 6.5%) when only taking face-to-face contacts into account (Fig. 3). --> I find it hard to find these numbers in the figure
NB: below the figure it states "numbers", on the y-axis "proportion"
Discussion:
Second sentence: We see a marked decrease of antibiotic prescriptions after the start of the first wave, only for a small part compensated by telephone consultations --> how can TCs compensate prescriptions?
In the summer of 2020 we see a partial recovery of antibiotic prescribing during OOH care followed by a second decline during the second wave --> is possible to add data from the end of the second wave to see whether is goes up again?
Author Response
Thank you for the opportunity to review our manuscript and valuable comments.
General remark all reviewers:
We noticed a potential source of confusion in the paper: the percentage decrease for the number of antibiotics prescribed corresponds to a relative decrease (i.e., takes into account the pre-lockdown number). However, the percentage decrease for the number of antibiotics prescribed per consultation corresponded to an absolute decrease in the previous draft of the manuscript. Since this may be confusing, we renamed this 'percentage points' and also added the relative decrease as a percentage. This also indicates that the contribution of the change in prescribing behavior is larger than how it was presented in the conclusion. Please find our changes:
In the abstract on lines: 30,31,36-41
In the text on lines: 154-160; 193-196, 352-356
Reviewer 1
Comments and Suggestions for Authors
The present paper presents the course of antibiotic prescribing over time in OOH GP-care in Belgium (2019-2020). Changes in numbers of prescriptions (total, amox, augm) are presented. The trend in prescription before-after lockdown is analysed using ITS. Data for nitrofurantion is analysed separately. A large decrease in prescription after lockdown is found.
The manuscript is clear, relevant for the field and presented in a well-structured manner.
The study design is appropriate.
General:
It would be valuable to see not only the numbers of antibiotic prescriptions but also the course of the number of contacts. These numbers are needed to support the conclusion of the paper (the COVID-19 pandemic has led to an unprecedented drop in the number of antibiotic prescriptions in Belgian OOH primary care, which is mostly explained by a decrease in the number of patient contacts after the implementation of the lockdown)
Response: We agree with the reviewer that including the number of contacts may clarify our findings. Because the study population included in 2020 was larger than in 2019, we choice to show the contact per weekend per 100000 population. We added following table.
|
Before lockdown |
After lockdown |
Number of contacts per weekend per 100 000 population (SD) |
|
|
- All |
121 (15) |
159 (39) |
- Face-to-face |
120 (15) |
85 (17) |
- Telephone |
1 (0) |
75 (29) |
Number of antibiotic prescriptions per weekend per 100 000 population (SD) |
|
|
- All (J01) |
26 (3) |
15 (3) |
- Amoxicillin (J01CA04) |
11 (2) |
4 (1) |
- Amoxicillin/clavulanate (J01CR02) |
6 (1) |
4 (1) |
SD: Standard deviation; J01, J01CA04, J01CR02: Anatomical Therapeutic Chemical codes
We also specified the contribution of face-to-face prescribing behavior in the text on lines: 154-160; 193-196, 352-356
With telephone contacts included, the total number of contacts increased, likely due to the accessibility of the telephone contacts.
With respect to the second part of the conclusion: GPs also prescribed fewer antibiotics per face-to-face patient contact, but this decrease was much
smaller: I cannot find the decrease over time for TCs in a figure. And how can GPs prescribe less than before COVID if before COVID there was no option for TC?
Response: Indeed, TC were not integrated as common practice before the COVID -19 lockdown. We noticed a relatively high number of TC when this possibility was formally implemented (i.e., after implementation of lockdown).The statement the reviewer is referring to relates to face-to-face prescribing behavior, this was indeed less. To clarify this statement and to explain the contribution of face-to-face prescribing behavior we amended the text on lines: 154-160; 193-196, 352-356
The option of prescribing an antibiotic during a telephone consultation was used very scarcely. In figure 1 we show the part of TCs were an antibiotic prescription was issued.
This was also discussed on lines 181 of the discussion section: "Only a limited number of antibiotic prescriptions were delivered during telephone contacts"
Did you look at the course of contacts for complications before/after lockdown? (pneumonia/mastoiditis/pyelonefritis). Do we have an idea whether the decrease in AB was safe?
Response: This is a very valuable remark. We do not have an idea if there were more complications. One could expect an increase in complications, but most likely during the days following after the OOH contact, during a visit with the own GP or for example a visit to the ED. There are no data of this type of contacts in our database. Also GPs changed their way of coding for diagnoses before and after the lockdown, for example a suspected pneumonia was coded as a pneumonia before and often as a suspected covid afterwards.
We chose not to add this information to the manuscript, because we can’t link this to the (un) safety of the decrease in antibiotic prescribing.
In the limitation section we stated: “We did not assess the appropriateness of prescribing and it is unclear whether there was, for example, an under treatment of infections or a rise in complications.” (line 230)
Introduction:
Were there differences in preventive measures between the first and second wave? Does this help explain the difference in decrease first/second wave?
Response: There were indeed differences in measures between the first and the second wave. We added this to the section 4.1 Study design and setting (line276): The first lockdown was a strict lockdown with, amongst other restrictions, closure of all schools, closure of non-essential shops, travel ban and mandatory teleworking. While for example in the second lockdown school contacts were only partially limited, travelling was limited but non-essential shops closed again and teleworking was obliged when possible.
It is difficult to make statements about the exact effect of these different measures. All measures however contributed to limiting the circulation of viral pathogens, and limiting physical contacts, included physical contacts with OOH primary care.
We added to the manuscript that further qualitative research is recommended to reveal why GPs' prescribing behavior has decreased.
Results
Could data be added on UTI contacts/nitrofurantoin per UTI contact? Nitro- numbers stayed the same: did number of UTI contact go down? And the prescription rate go up?
Response: Thank you for this question. We added this information in a figure as supplementary material. The contacts with a UTI diagnosis per population of 100 000 stayed stable before and after the lockdown as did the prescriptions of nitrofurantoin for a UTI diagnosis per population of 100 000. And referred to this figure in 2.2.4 trends in prescribing nitrofurantoin. (line 161)
The figures show a clear picture over time. The numbers in the figures are highly variable over time. Could a line with a mean before/after lockdown be added. So the overall decline numbers can be visualised?
Response: we added a line at the start of the lockdown to all figures. And we added table 1 with the number of antibiotic prescriptions and the number of contacts per weekend per 100 000 with the standard deviation.
i.e. Figure three: text: Antibiotic prescribing per contact, both face-to-face and telephone contacts, de creased by 12.0% ( 95% CI: 10.8 – 13.3%) while it decreased by 5.2 % (3.8 – 6.5%) when only taking face-to-face contacts into account (Fig. 3). --> I find it hard to find these numbers in the figure
Response: The title of Figure 3 indeed was not correct and the way it was referred to in the text was confusing. we only show the proportion of prescriptions per face-to-face contact per weekend in the figure. We corrected this on line 163.
We also added an extra table to present the number of antibiotic prescriptions and contacts, used to show the proportions of antibiotic prescribing per contact, before and after the start of the lockdown, both for all contacts and for face-to-face contacts only.
NB: below the figure it states "numbers", on the y-axis "proportion"
Response: Thank you for this comment. We changed the text below this figure.
Discussion:
Second sentence: We see a marked decrease of antibiotic prescriptions after the start of the first wave, only for a small part compensated by telephone consultations --> how can TCs compensate prescriptions?
Response: This sentence could do with some more clarification. During the lockdown, for the first time in Belgium, the possibility was organized to issue a prescription by telephone. We changed the sentences in line 139-140 and 182-184
We see a marked decrease of antibiotic prescriptions for RTI-related face to face contacts after the start of the first wave, only for a small part compensated by antibiotic prescriptions issued during telephone consultations.
In the summer of 2020 we see a partial recovery of antibiotic prescribing during OOH care followed by a second decline during the second wave --> is possible to add data from the end of the second wave to see whether is goes up again?
Response: We agree that it would have been interesting to explore what happened in the period proposed by the reviewer. However, based on the study aim the specific study period until December 2020 was defined. Data request and ethical permission were sought accordingly.
Thank you for your time, and thorough review and providing us the opportunity to strengthen our research.
Reviewer 2 Report
In the manuscript entitled " Antibiotic prescribing trends in Belgian out-of-hours primary
care during the COVID-19 pandemic: observational study using routinely collected health data". The topic is great. The studies reported in this manuscript fit the aims and scope of the journal and need some revision.
write the keywords according to Mesh and alphabetic order
How did you calculated sample size? Based on already published articles or did you perform power analysis?
Write the gender and age of patients
the method is poor designed. Exclusion and inclusion criteria, sample collection, etc., should be mentioned..
Author Response
Thank you for the opportunity to review our manuscript and valuable comments.
General remark to all reviewers:
We noticed a potential source of confusion in the paper: the percentage decrease for the number of antibiotics prescribed corresponds to a relative decrease (i.e., takes into account the pre-lockdown number). However, the percentage decrease for the number of antibiotics prescribed per consultation corresponded to an absolute decrease in the previous draft of the manuscript. Since this may be confusing, we renamed this 'percentage points' and also added the relative decrease as a percentage. This also indicates that the contribution of the change in prescribing behavior is larger than how it was presented in the conclusion. Please find our changes:
In the abstract on lines: 30,31,36-41
In the text on lines: 154-160; 193-196, 352-356
Reviewer 2:
Comments and Suggestions for Authors
In the manuscript entitled " Antibiotic prescribing trends in Belgian out-of-hours primary care during the COVID-19 pandemic: observational study using routinely collected health data". The topic is great. The studies reported in this manuscript fit the aims and scope of the journal and need some revision.
write the keywords according to Mesh and alphabetic order
Response: Thanks you for this suggestion. We changed them to alphabetic order and used MeSH terms.
How did you calculated sample size? Based on already published articles or did you perform power analysis?
We agree that sample size calculation is important, to avoid the study being underpowered, but also when data collection is challenging. These reasons did not apply to our case. We did not use a sample but the entire available database. We did expect to find an effect because of the large amount of available secondary data and because of the relatively large anticipated effect size which we estimated based on field experience. The question was rather to measure the decrease in prescriptions as precisely as possible and secondary, to explore any potential reasons for this decrease by analyzing trends in specific groups of antibiotics. Therefore, we did not perform a formal sample size calculation for this study. Given the large numbers of contacts and prescriptions, the issue is indeed not a potential lack of power, but rather the potential of statistically significant findings having little clinical relevance.
Decreases in the proportion of antibiotic prescribing as small as 5 percentage points (absolute differences of 5%) are clinically relevant.
Write the gender and age of patients
Response: In the result section we wrote: Patients' average age was 34.0 years (SD 23.5), 54.1% were female. The study population before the start of the first lockdown (13/03/2020) was older (mean 36 year) and included more females (54.3%)) compared to the population after implementation of the lockdown (mean 31 years, 53,8% female), which is only a very small difference and therefore not relevant.
the method is poor designed. Exclusion and inclusion criteria, sample collection, etc., should be mentioned.
Response: Thank you for this comment. We did not explicitly state exclusion or inclusion criteria as there were no patient-related exclusion or inclusion criteria. All contacts with a GP for patients contacting the participating GPC during weekends and holidays (from here onwards we refer to weekends and bank holidays as weekends) in 2019 and 2020 were included, as stated in the methods section under 4.3.. Exclusion of data related to the data collection process was discussed under 4.3. We further mentioned that data for this study were routinely collected using electronic health records in section 4.2. All the variables that were included in the database were mentioned in 4.4. No samples were collected for the purpose of this study.
Thank you for your time, and thorough review and providing us the opportunity to strengthen our research.
Reviewer 3 Report
The authors observed and evaluated statistically the number of antibiotic prescriptions in primary care at Belgium during the COVID pandemic. They found a dropping in antibiotics prescriptions which was explained as a decrease in patients contacts during the lockdown. Generalists doctors also prescribed less antibiotics. Decrease in antibiotics was more pronounced in Respiratory tract infections patients.
The paper is well written .results are well investigated and should be quite informative and interesting for readers .Results of this paper could eventually change the attitude of patients and doctors for the antibiotic policy.Bibliography is up to date.
My suggestion is to ACCEPT and publish it in its present form
Author Response
Thank you for the review and this kind feedback.
General remark to all reviewers:
We noticed a potential source of confusion in the paper: the percentage decrease for the number of antibiotics prescribed corresponds to a relative decrease (i.e., takes into account the pre-lockdown number). However, the percentage decrease for the number of antibiotics prescribed per consultation corresponded to an absolute decrease in the previous draft of the manuscript. Since this may be confusing, we renamed this 'percentage points' and also added the relative decrease as a percentage. This also indicates that the contribution of the change in prescribing behavior is larger than how it was presented in the conclusion. Please find our changes:
In the abstract on lines: 30,31,36-41
In the text on lines: 154-160; 193-196, 352-356
Reviewer 4 Report
The authors described the trends observed in the antibiotic prescribing for RTIs before and after COVID-19 lockdown in Belgium using existing electronic record. However, there are several major issues that need too be addressed as highlighted below.
Major comments:
- The aim of the study is not clear. From the layout of the manuscript, I’m not sure if the authors analyze all the data for RTIs only (Line 80) or all the prescribing data in the population. Out of sudden, the nitrofurantoin which is the first antibiotic choice for UTI is presented as well. The title is also ambiguous.
- Only two antibiotics for RTIs were included for analysis which may be underrepresented.
- The age group and gender of the population should be taken consideration during the analysis.
- Line 140, when is the second wave? Please specify the date and is it included in the analysis? The observation needs to be supported by data.
- The results of this study do not provide new insights into the prescribing trends due to the limited number of variables and data used for analysis. All the results are anticipated due to lockdown.
Specific comments:
- Line 85, provide specific month of the data used for analysis.
- The trends in all the figures will be better represented in line chart with the prescriptions grouped by month.
- Figure 1, indicate in the graph when the lockdown occurs. Same for other figures.
- Comparisons of Figure 2 and 4 can be done in a single line chart.
Author Response
Thank you for the opportunity to review our manuscript and valuable comments.
General remark all reviewers:
We noticed a potential source of confusion in the paper: the percentage decrease for the number of antibiotics prescribed corresponds to a relative decrease (i.e., takes into account the pre-lockdown number). However, the percentage decrease for the number of antibiotics prescribed per consultation corresponded to an absolute decrease in the previous draft of the manuscript. Since this may be confusing, we renamed this 'percentage points' and also added the relative decrease as a percentage. This also indicates that the contribution of the change in prescribing behavior is larger than how it was presented in the conclusion. Please find our changes:
In the abstract on lines: 30,31,36-41
In the text on lines: 154-160; 193-196, 352-356
Reviewer 4
The authors described the trends observed in the antibiotic prescribing for RTIs before and after COVID-19 lockdown in Belgium using existing electronic record. However, there are several major issues that need too be addressed as highlighted below.
Major comments:
The aim of the study is not clear. From the layout of the manuscript, I’m not sure if the authors analyze all the data for RTIs only (Line 80) or all the prescribing data in the population. Out of sudden, the nitrofurantoin which is the first antibiotic choice for UTI is presented as well. The title is also ambiguous.
Only two antibiotics for RTIs were included for analysis which may be underrepresented.
Response: Thank you for this comment. To show the effect of the pandemic on antibiotic prescribing, which is the main aim of our study, we have analyzed all the prescribing data in the population and shown the trend in prescriptions of all antibiotics (ATC code J01) per 100 000 inhabitants. Among all antibiotics prescribed and consumed in Belgium, amoxicillin and amoxicillin/clavulanate are used most often and their main indications are respiratory tract infections (RTIs). Following reference conforms this statement: Colliers et al, Antibiotic prescribing quality in out-of-hours primary care and critical appraisal of disease-specific quality indicators, Antibiotics 8.2 (2019): 79. Hence we used prescribing of amoxicillin and amoxicillin/clavulanate as a proxy to assess antibiotic prescribing for RTIs, while we used prescribing of nitrofurantoin, the first choice antibiotic for urinary tract infections, to assess trends in prescribing for other infections than RTI. In addition, we have provided the trends in prescriptions of all antibiotics for possible RTI-related and RTI-unrelated contacts as supplemental material in Figurse S1 and S2.
The age group and gender of the population should be taken consideration during the analysis.
Response: Thank you for this suggestions. The analysis was repeated with the suggested covariates included. This did not result in any major changes. All estimates were modified accordingly throughout the manuscript. Our findings and conclusion remained the same.
Line 140, when is the second wave? Please specify the date and is it included in the analysis?
Response: This information is provided in the Methods section, which can be found after the Result section, in line with the journal's article format. In the Methods section, we stated: In Belgium the first wave of the COVID-19 pandemic occurred between March 1 and June 21, 2020 (with a peak in week 15), and the second wave between August 31, 2020 and February 14, 2021 (with a peak in week 45-46).
The end of the second wave was not included in the analysis as this period was not part of the primary aim of the study. Based on the study aim the specific study period until December 2020 was defined. Data request and ethical permission were sought accordingly.
The observation needs to be supported by data.
The results of this study do not provide new insights into the prescribing trends due to the limited number of variables and data used for analysis. All the results are anticipated due to lockdown.
Response: We respectfully disagree with the reviewer. The aim of our study was to describe actual trends in antibiotic prescribing in OOH primary care. In addition, we explored potential reasons for these trends. Therefore, we also analysed the trends of relevant antibiotic substances, the number of contacts and the number of antibiotic prescriptions per contact type. In these analyses, we included 388 293 contacts and 268 430 prescriptions, over two consecutive years, covering a population of 3 162 345 inhabitants. Upon your suggestion, we have also repeated our analysis taking into account information on age and gender of the population. We have estimated that the COVID-19 related lockdowns had a substantial effect on not only the total number of antibiotic prescriptions, but also on the proportion of antibiotic prescriptions per (physical) contact in OOH primary care. In addition, we have discussed multiple hypotheses for this effect, such as triage of patients with the possibility of postponement of care, the discouragement of patients to contact a GP for mild RTIs, more telephone consultations, hygiene measures and the social distancing measures. Although more in-depth research is indeed needed to better understand what happened and what the long-lasting effects will be of COVID-19 on antibiotic prescribing and the consultation of patients with RTIs, from the trends in antibiotic prescribing and possible explanations we have described valuable lessons can be learned for the future of antimicrobial stewardship both for clinicians, policymakers and fellow researchers.
Specific comments:
Line 85, provide specific month of the data used for analysis.
The trends in all the figures will be better represented in line chart with the prescriptions grouped by month.
Response: Thank you this suggestion, however we respectfully disagree. The lockdown started in mid-March, so if you look at it month by month it would look like a gradual transition from February to April, when it is more of a radical breakpoint. We also think the natural variation per weekend is relevant to show.
Figure 1, indicate in the graph when the lockdown occurs. Same for other figures.
Response: Thank you for this suggestion. Indicating in the figures when the lockdown took place indeed improves the clarity and was therefore implemented in revised version of these figures.
Comparisons of Figure 2 and 4 can be done in a single line chart.
Response: Thank you for this suggestion. We adjusted the figures 2 until 4 with a single line.
Thank you for your time, and thorough review and providing us the opportunity to strengthen our research.
Round 2
Reviewer 1 Report
Thank you for the answers and revisions to the manuscript.
Reviewer 4 Report
Dear authors,
Line 27, some units are missing. 11.47 refers to what? Same issues found throughout the manuscript.